# Pediatric Crohn’s Disease in the Upper Gastrointestinal Tract: Clinical, Laboratory, Endoscopic, and Histopathological Analysis

**DOI:** 10.3390/diagnostics14090877

**Published:** 2024-04-24

**Authors:** Dunja Putniković, Jovan Jevtić, Nina Ristić, Ivan D. Milovanovich, Miloš Đuknić, Milica Radusinović, Nevena Popovac, Irena Đorđić, Zoran Leković, Radmila Janković

**Affiliations:** 1Institute of Pathology “Prof. dr Đorđe Joannović”, Faculty of Medicine, University of Belgrade, 11000 Belgrade, Serbia; lordstark90@gmail.com (J.J.); milos.djuknic@med.bg.ac.rs (M.Đ.); 2Department of Gastroenterology, University’s Children’s Hospital, 11000 Belgrade, Serbia; ninaristic13@gmail.com (N.R.); imilovanovich@gmail.com (I.D.M.); milicaradusinovic@hotmail.com (M.R.); nevena.popovac@gmail.com (N.P.); djordjicirena@gmail.com (I.Đ.); zlekovic2000@yahoo.com (Z.L.); 3Faculty of Medicine, University of Belgrade, 11000 Belgrade, Serbia

**Keywords:** Crohn’s, granuloma, endoscopy, esophagogastroduodenoscopy, colonoscopy, histopathology

## Abstract

Crohn’s disease (CD) is a progressive, multifactorial, immune-mediated disease characterized by chronic inflammation of any part of the gastrointestinal (GI) tract. Pediatric patients present with a more extensive form of the disease, especially in the upper GI tract with various histopathological inflammatory patterns. Our study aims to analyze the clinical, laboratory, endoscopic, and histopathological findings in children with diagnosed CD and compare results on the initial and follow-up tests. We have included 100 children and adolescents with CD, with performed endoscopic and histopathological (HP) procedures. The results of multiple biopsies executed in these 8 years were matched and compared. We found a statistically significant frequency reduction in stool changes (65.52% to 18.18%), weight loss (35.24% to 4%), and abdominal pain (41.86% to 6.67%) as presenting symptoms. There was an improvement in all laboratory values: fecal calprotectin (1000 to 60,8 μg/g), C-reactive protein (12.2 to 1.9 mg/L), and albumin (36 to 41 g/L). On esophagogastroduodenoscopy and ileo-colonoscopy 36.59% and 64.86% patients had specific findings, respectively. A total of 32 patients had evidence of Crohn’s disease in the upper GI tract. Non-caseating granulomas were found on 9% of oesophageal, 18% of gastric, and 12% of duodenal biopsies. In the lower GI tract, we have observed a disease progression in the rectum (72.29 to 82.22%) and descending colon (73.49 to 80%). There was no registered disease progression in the upper GI tract. Our study demonstrated a significant decline in the frequency of symptoms and an improvement in laboratory values on the follow-up examinations. More than a third of our patients had specific endoscopic and HP findings in the upper GI tract, and an additional 23% had HP findings highly suggestive of CD. We demonstrated the importance of regular clinical, laboratory, endoscopic, and histopathological assessments of pediatric CD patients.

## 1. Introduction

Crohn’s disease (CD) is a progressive, multifactorial, immune-mediated disease characterized by chronic inflammation in any part of the gastrointestinal (GI) tract [1,2,3]. It is thought to occur due to an altered immune response to unknown environmental factors in a genetically predisposed person, with a peak age of onset between 15 and 30 years of age [3,4,5]. The incidence of pediatric-onset CD is rising globally and varies geographically with a recorded incidence in Europe of 0.2–23 per 100,000 people [5,6,7]. Pediatric patients present with a more extensive form of the disease, accompanied by non-specific gastrointestinal symptoms and weight loss, causing a concerning delay in the linear growth of immature children [2,5]. The risk of having a disabling condition 5 years after diagnosis is higher in pediatric patients and is associated with the psychosocial effects of living with a chronic incurable disease [1,6]. The diagnosis of CD is complex and depends on the combination of clinical signs and symptoms, laboratory (fecal calprotectin, C-reactive protein, albumin, and others), and endoscopic and histopathological data. These parameters are used in disease monitoring, as multiple grading scores are developed but not yet part of a standardized practice, especially in pediatric patients [2,3,8]. Although it can arise anywhere from the mouth to the anus, a great proportion of children have the burden of active disease in the upper GI tract with various inflammatory patterns on endoscopy and histopathology (HP) [5]. According to standard criteria, an esophagogastroduodenoscopy (EGDS) with biopsy obtainment is performed in all children at initial evaluation, in addition to the ileo-colonoscopy. On HP analysis, chronic inflammation with non-caseating granulomas is a characteristic feature to define CD. Still, other findings can be highly suggestive of the presence of this condition, such as focally enhanced gastritis and lymphocytic esophagitis [9]. Standardized histological grading systems have not yet been developed for pediatric CD [10]. Upon the completion of the assessment of disease severity and activity, a prediction of progression can be made, and aid in the choice of personalized therapy for every individual patient [1]. Because of these variations in the presentation of the disease, our study aims to identify the fraction of specific and highly suggestive findings of CD on the upper endoscopy and histopathology in children with CD. We will also compare the results of the initial and follow-up results in patients who had more than one examination at this time, with a correlation between endoscopic and histopathologic findings with the clinical and laboratory presentation of our pediatric patients.

## 2. Materials and Methods

Our retrospective study included 100 children and adolescents with Crohn’s disease treated at the University Children’s Hospital in Belgrade, Serbia, in the period between January 2016 and December 2023, where the EGDS and ileo-colonoscopy with biopsy was performed. The histopathological samples were analyzed at the Institute of Pathology ‘‘Prof. dr Đorđe Joannović’’, Faculty of Medicine, University of Belgrade. All children who underwent upper and lower endoscopy with biopsy obtainment and histopathological analysis with confirmed CD were included in this study. Patients with non-classified colitis, ulcerous colitis, or suspected CD were not a part of this study. During this period, 68 patients were diagnosed with CD for the first time and 32 patients had already been diagnosed in the past and had a follow-up biopsy. A total of 45 patients had two biopsies during these 8 years, and all their results were matched and compared. From the medical documentation, patient demographics (age at diagnosis, first and second biopsies, and gender), duration of symptoms before hospitalization, treatment modality, and laboratory values (fecal calprotectin, C-reactive protein, and albumin) were collected. The referent values for these parameters were standardized with the biochemical laboratory of the University Children’s Hospital: fecal calprotectin (FC) < 60 μg/g, C-reactive protein (CRP) < 3 mg/L, and albumin 38–56 g/L. Four symptoms were recorded during the evaluation: stool changes (frequent bowel movements or diarrhea), abdominal pain, anemia, and weight loss. The patients’ previous drug regimens were mentioned but not quantified. We described the findings of upper endoscopy and ileo-colonoscopy and classified them as specific (which included either of the following: cobblestone-like mucosa, aphthous lesions, mucosal ulcers, erosions, and strictures), non-specific (edema, hyperemia), and normal. We noted their localization in patients with specific upper endoscopic changes (esophagus, stomach, and duodenum). At the time of endoscopy, the biopsy set consisted of samples taken from the esophagus, stomach, duodenum, terminal ileum, cecum, ascending colon, transverse colon, descending colon, sigmoid colon, and rectum. The slides were stained with hematoxylin and eosin [H&E] and evaluated by a pathology specialist. The HP findings were further classified appropriately, with an emphasis on those that were specific (chronic inflammation with non-caseous granulomas) and highly suggestive of CD (focally enhanced gastritis and lymphocytic esophagitis). Regarding biopsies taken during ileo-colonoscopy, we noted whether signs of CD were detected and their localization and consequently classified the findings based on the Paris classification. The Paris classification for CD is comprised of 4 parameters: age of onset, localization, phenotype, and growth. The age of onset is divided into three groups: <10 (A1a), 10–16 (A1b), and 17–39 (A2). The localization of the disease is divided into the involvement of the distal third of the ileum, with or without limited cecal disease (L1), colonic (L2), ileocolonic (L3), upper disease proximal (L4a), and distal (L4b) to the ligament of Treitz. Patients with L1-3 disease localization can have a coexisting L4 involvement. The phenotype can be inflammatory (B1), stricturing (B2), penetrating (B3), and both stricturing and penetrating (B4). Growth delay is categorized as non-existent (G0) and present (G1). We also noted the Global Histologic Disease Activity Score (GHAS) for the terminal ileum and colon, which was recorded independently by a pathology specialist. It consists of 8 items, scored individually for the terminal ileum and colon. The score range was from 0, which indicated no activity, to 14, which represented severe disease activity. Furthermore, a value of ≤4 implied mild, 5–9 moderate, and ≥10 severe disease activity. We analyzed disease activity progression in patients on different treatment modalities. Statistical analysis was performed in *EZR* (R package version 0.1.4). We have compared nominal variables using the Chi-squared (ꭓ^2^) test for independent, and McNemar’s test for dependent variables. Numerical and ordinal variables were evaluated using a non-parametric Wilcoxon’s signed rank test. The results were represented as medians and 1st–3rd quartiles (Q1–Q3). Statistical significance was set at *p* < 0.05.

## 3. Results

The median age of patients was 13 years and 8 months at the time of diagnosis, and 15 years and 7 months on the second biopsy. Five of our patients can be classified into the very early onset inflammatory bowel disease (VEO-IBD) category, i.e., patients diagnosed before the age of 6. The male-to-female ratio was 1.68 (62 boys and 38 girls). The median duration of symptoms before diagnosis was 5 weeks (0.2–60 weeks). The median time between the initial and first follow-up biopsy was 1 year and 3 months (1 month to 2 years and 4 months). The most frequent presenting symptoms were stool changes (65.52%), followed by abdominal pain (41.86%), weight loss (35.23%), and anemia (11.49%). We compared the frequencies of symptoms in time of the first and second biopsy obtention which are shown in Figure 1. A statistically significant decrease in anamnestic data regarding stool changes (*p* < 0.001), weight loss (*p* < 0.001), and abdominal pain (*p* < 0.001) was observed.

Immunomodulators were the most common drug modality (54) of our patients, followed by biologic therapy (44) and corticosteroids (prednisolon for induction short-term and rarely budesonid for long-term therapy; 27). A smaller proportion of patients were treated with other types of medications (anti-inflammatories: 12, proton pump inhibitors: 10, antibiotics: 14).

At the time of the first visit, 86/89 (96.63%) patients had elevated FC levels, and at the time of the second 22/44 (50%). Its median value was 1000 μg/g (26.17–10,000 μg/g) in the first, and 60.8 μg/g (0–2531 μg/g) in the second examination. On the first and second visits, 72/93 (77.42%) and 15/51 (29.42%) patients had elevated CRP, respectively. The median value of the first CRP was 12.2 mg/L (0.6–188.8 mg/L) and the second was 1.9 mg/L (0–109.5 mg/L). On the first and second visits, 58/91 (56%) and 10/90 (11.11%) patients had decreased albumin levels, respectively. The median albumin value was 36 g/L (20–44 g/L) and 41 g/L (18–46 g/L) in the first and second visits, respectively. All three laboratory parameters improved significantly between the first and second patient visit, with decreased FC and CRP and increased albumin levels (Wilcoxon’s test, *p* < 0.001). As patients’ therapy modality could influence this statistical difference, a therapy/laboratory test was performed (Wilcoxon’s test). Patients treated with biologics had a statistically relevant change in all three parameters (*p* < 0.001; medial difference: FC = 380.35, CRP = 8.7, albumin = 5), as did patients treated with immunomodulators (*p* < 0.001 for FC and *p* < 0.05 for CRP and albumin; medial difference: FC = 962.4, CRP = 2, albumin = 3). Patients treated with corticosteroids had a significant decrease in FC levels (*p* < 0.05; median difference = 949.4), but not in CRP (*p* = 0.231) and albumin (*p* = 0.238).

We analyzed endoscopic findings at the upper and lower endoscopy in patients with CD. On EGDS, 32 (36.59%) patients had specific, 20 (24.39%) had non-specific findings, and 30 (36.59%) were normal. Of patients who had specific findings, 50% were in the stomach, 28.57% in the duodenum, and 21.43% in the esophagus. On ileo-colonoscopy, 48 (64.86%) patients had specific, 16 (21.62%) had non-specific findings and 10 (13.51%) were normal. We have compared and presented the simultaneous findings on both endoscopies (ꭓ^2^ = 9.1, *p* = 0.059) in Table 1. Nineteen patients had specific changes on both colonoscopy and EGDS. Typical endoscopic findings suggestive of CD in the proximal GI tract are shown in Figure 2. We analyzed whether therapy choice had an impact on the endoscopic findings. Our study found significant improvement in patients receiving biologics, as they had a higher percentage of normal (52.5%/21.4%) and a lower percentage of specific (22%/52.4%) and non-specific (22.5%/26.2%) endoscopic changes (*p* < 0.05). Also, a borderline significant change (*p* = 0.05) was observed in patients on immunomodulators, as they had a higher incidence of normal (42.2%/30.6%) and non-specific (31.1%/16.7%) and a lower percentage of specific (26.7%/52.8%) findings on upper endoscopy. Therapy modality did not influence a change in endoscopic results in the ileo-colonoscopy (*p* > 0.05).

We further analyzed the histopathological profile of each biopsy. The diagnosis of CD was noted with the presence of chronic inflammation with non-caseous granulomas. A total of 31 patients had characteristic signs of CD in the first biopsy taken from the upper GI tract: 13 in the stomach, 5 in the esophagus, 6 in the duodenum, 2 in the esophagus and duodenum, 2 in the stomach and esophagus, and 3 in the stomach and duodenum. Only 2 patients had specific HP findings in the second examination, one in the stomach and one in the esophagus. Highly suggestive findings alone had 23 patients (17 focally enhanced gastritis and 6 lymphocytic esophagitis) (Figure 3).

The pathohistological findings on the first gastric biopsy are presented in Figure 4. Signs of CD were found in 18 patients (18%) and a normal gastric wall had 30 (30%). Focally enhanced gastritis (FEG) had 25 patients, active chronic gastritis (ACG) 5, inactive chronic gastritis (ICG) 18, and an elevated number of eosinophils in the gastric mucosa (EoG) 2 of our patients.

On the second gastric biopsy, 1 (2.22%) patient had signs of CD, 22 (48.89%) had normal findings, 8 (17.78%) FEG, 13 (28.89%) ICG, and 1 (2.22%) EoG. There was no statistically significant difference between the first and second biopsies (McNemar’s test, *p* = 0.178).

Specific signs of CD were found in 9 (9%) esophageal and 12 (12%) duodenal biopsies. A normal biopsy had 52 patients in their esophagus and 37 in the duodenum. In the esophagus, signs of gastroesophageal reflux disease were seen in 11 patients, lymphocytic esophagitis in 6, and eosinophilic esophagitis in 2 patients. Chronic duodenitis was recorded in 48 and an elevated number of eosinophils in the duodenal mucosa in 1 patient. In twenty patients a biopsy of the esophagus was not obtained and in 2 of the duodenum (Figure 5). One patient had the presence of CD on the esophagus in the second biopsy. No statistically significant changes were observed between the first and second biopsies in the esophagus or duodenum.

In Figure 6, we have compared the findings between the first and second biopsy results for each segment of the lower GI tract and found that the signs of CD were significantly increased in the second biopsy in the descending colon (*p* = 0.02) and rectum (*p* = 0.035). On the first biopsy, 73.49% of patients had signs of CD in the descending colon and 72.29% in the rectum. On the second biopsy, 80% of patients had signs of CD in the descending colon and 82.22% in the rectum. Of the patients diagnosed for the first time with CD, 68% had specific findings in the entire lower GI tract.

The Paris classification for Crohn’s disease and GHAS on the initial examination are shown in Table 2. No statistically relevant progression of the disease to a higher L category (Wilcoxon’s test, *p* = 0.574) was observed, even when stratifying patients based on the therapy modality they were receiving (biologics *p* = 0.12, corticosteroids *p* = 0.285, immunomodulators *p* = 0.337). However, there was a significant increase in disease activity from mild to moderate (GHAS) in patients on biologics in both the colon and terminal ileum and in patients on immunomodulators in the terminal ileum (Mann–Whitney U test, *p* < 0.05).

## 4. Discussion

Inflammatory bowel disease (IBD) can occur at any age, but almost a third of patients are diagnosed in the period of childhood or adolescence. The diagnosis of this condition in children is different from the adult forms, as they are often diagnosed before the development of disease complications [2,6]. In our study, the median age of diagnosis was 13 years and 8 months, and the disease was more frequent in males (62%), both characteristics consistent with older-onset IBD in children [10]. The protocol in our hospital for children with CD is that upon diagnosis follow-up visits should be scheduled in 6 to 9-month intervals. Because data in this study involves a period of the COVID-19 pandemic, the median time between the two examinations was 1 year and 3 months (1 month to 2 years and 4 months). The duration of signs and symptoms before diagnosis was 5 weeks (to a maximum of 5 months) which is less than in a Swiss IBD cohort which found the diagnostic delay of an average of 3 months in pediatric patients [11]. The dissimilarities in presentation in CD are one of the reasons for this diagnostic delay, but most commonly involves non-specific gastrointestinal symptoms such as abdominal pain, diarrhea, nausea, vomiting, or hematochezia [2,4]. Our study showed that stool changes were the most common presenting symptoms (65.52%) and that they, along with abdominal pain and weight loss, significantly subsided at the time of a follow-up visit. Weight loss is a particularly concerning clinical sign because it can lead to a defect in linear growth, which is seen in CD in up to 40% of pediatric patients. This is because of the effects of chronic gastrointestinal inflammation accompanied by malabsorption, increased metabolic demand, poor oral intake, and the adverse effects of corticosteroids and immunomodulators. Along with clinical signs, laboratory values are used for aiding the diagnosis of CD, but even so, up to 20% of children have normal marker levels [2]. Fecal calprotectin, a cytosolic protein released by neutrophils, is a stool marker of intestinal inflammation used for the assessment of disease activity and progression and is a screening and exclusion tool for suspected IBD patients when values exceed 50–100 μg/g and ≤40 μg/g, respectively [12,13]. In our study, at the index visit 86 out of 89 patients had elevated values of FC, with a median value of 1000 μg/g which significantly declined in time of the second visit (60.8 μg/g), similar to other studies [3]. CRP, an acute-phase protein, is frequently elevated in pediatric patients and has also been shown to aid in the diagnosis and prognosis of CD because it correlates well with the degree of the inflammatory process seen in this condition [1,14]. In our study, 77.4% of patients had elevated CRP and the median value significantly declined from 12.2 to 1.9 mg/L between the two visits, which is similar to data in the literature [3]. The concentration of serum albumin indicates the patient’s nutritional status, the presence of systemic inflammation, and low values often signal poor prognosis [14]. Hypoalbuminemia on the index and follow-up visits was registered in 56% and 11.11% of patients, respectively. Although CRP and albumin are used in the evaluation of CD, they are not specific and do not correlate well with small-bowel CD which should be mentioned [1]. For the small-bowel, CD, a magnetic resonance enterography is used [2].

Endoscopy is a gold standard for the diagnosis and evaluation of the disease activity, as EGDS and ileo-colonoscopy enable the macroscopic evaluation of the intestinal mucosa and histopathological biopsy analysis [2,3]. A total of 19 patients had specific changes on both the upper and lower endoscopies. Specific findings on EGDS and ileo-colonoscopy had 36.6% and 64.9% of patients, respectively. The percentage of our patients with these findings on upper endoscopy was larger than the European CD registries report (9–24%), which shows that a properly performed EGDS is a good diagnostic tool for CD [15,16]. Abuquteish et al. [5] reported that the most common site for endoscopic deviations is in the stomach, followed by the esophagus and duodenum. In our study, the stomach was the site of the most frequent macroscopic changes (50%) as well, followed by the duodenum (28.57%) and esophagus (21.43%). This shows the importance of routine endoscopic procedures in the diagnosis and monitoring of CD, as it can serve in the assessment of disease progression and suggest underlying pathology.

With further evaluation of the biopsies obtained during EGDS, diagnostic signs of CD (chronic inflammation with non-caseating granulomas) were found in 18% of patients in the stomach, 9% in the esophagus, and 12% in the duodenum. Compared with the EUROKIDS registry [17], the incidence of granulomas in our study was higher in all segments of the upper GI tract: the stomach (11.5%), esophagus (4.7%), and duodenum (3.3%). Histologically, non-caseating granulomas consist of 5 or more epithelioid histiocytes and/or multinucleated giant cells and are the distinguishing feature between CD and ulcerative colitis [5,10]. Histopathological findings suggestive, but not characteristic of CD are focally enhanced gastritis and lymphocytic esophagitis. FEG was seen in 25% of our patients in the first and 28.89% in the second biopsy, while the reported frequency in the literature is up to 50%. Children with this finding are 15 times more likely to be diagnosed with either CD or ulcerative colitis. It is defined as a focal pit inflammation comprised of lymphocytes and histiocytes most commonly seen in the gastric antrum. These patients are also more likely to have signs of CD somewhere else in the gastrointestinal tract, so further endoscopic and HP evaluation is mandatory [18,19]. Lymphocytic esophagitis, another histopathological finding highly suggestive of pediatric CD, is diagnosed when more than 20 intraepithelial lymphocytes are found in one high-power field (HPF), with no significant number of granulocytes. This diagnosis is not yet standardized, as lymphocytes are normally found in the esophageal mucosa, mainly in the peripapillary epithelium (10–12/HPF). In our study, the prevalence of LyE was 6% in the first and 6.7% in the second biopsy, which was similar to a study conducted by Sutton et al. [20]. In the duodenum, the most common non-specific finding seen in patients with CD is chronic duodenitis whose prevalence ranges from 33–48% [5]. In our study, chronic duodenitis was the most frequently seen finding on both the first (48%) and second (46.7%) duodenal biopsies. We also found it interesting that eosinophilic esophagitis was present in half (52%) and an elevated number of eosinophils in the duodenum in a third (37%) of our patients. Eosinophils play an important role in both pro-inflammatory and anti-inflammatory processes, depending on the extent of the infiltration, and in some cases indicate disease remission [10]. The variability of HP findings in our study shows unpredictability in the biopsy results in pediatric patients with CD, and the need to find a more standardized approach to the disease diagnosis and distinguish findings that are highly suggestive of CD and whose appearance prompts a more detailed assessment of the entire GI tract of affected patients to reach the diagnosis with minimal delay time.

A total of 32% of children had CD characteristics in the upper GI tract (L4a). However, the HP findings on the ileo-colonoscopy showed that 2/3 (68%) of patients had complete ileo-colonic disease (L3), and limited ileocecal (L1) and colonic (L2) disease had 17% and 8%, respectively. This study shows a predominance of complete ileo-colonic disease in children with CD. We have observed that CD changes worsened significantly in the rectum and descending colon between the first and second biopsies, which indicates a predominance of disease progression in the left colon. In the groups of patients diagnosed for the first time with CD, 68% had specific findings in the entire lower GI tract, which shows that the HP analysis of CD should always include both upper and lower GI tract biopsies.

The most commonly used index for the assessment of disease activity in the terminal ileum and colon is GHAS, which was frequently noted in pathology reports [21]. Our study demonstrated a median low score in the terminal ileum (GHAS = 3), and a high value in the colon (GHAS = 7). Many (28.89%) had no disease activity in the terminal ileum. In both the terminal ileum and colon moderate GHAS was the most common result (31.1% and 43.1%, respectively).

The treatment of CD depends on the extent and severity of the disease [1]. The main therapeutic goal in pediatric CD is achieving and maintaining clinical remission of the disease, as well as reaching mucosal healing, decreasing symptom occurrence, improving life quality, and preventing linear growth defects while keeping the adverse drug effects to a minimum [4]. The most common drug regimen used in our patients was immunomodulators (azathioprine or methotrexate), which is a common first-line therapy choice used for maintenance of CD remission. Methotrexate is an anti-folate agent while azathioprine is a purine analogue [7]. Biologics were the second most prevalent treatment, i.e., Infliximab and Adalimumab (chimeric monoclonal IgG antibodies to TNFα, a prominent gastrointestinal pro-inflammatory cytokine) both approved for use in pediatric patients with excellent success rates [4]. Our study also analyzed the impact of different therapy modalities on the laboratory, endoscopic, and histopathologic results. The improvement of laboratory parameters was unchanged in different drug regimens, except in patients treated with corticosteroids when no correction in CRP and albumin levels were observed. This can be explained due to the usage of corticosteroids for a short period (up to 3 months) for remission induction, and in severe cases. Similarly, it did not impact the upper and lower endoscopy results, with the exception of patients on biologics (more than half of patients had a normal endoscopy) and immunomodulators (lower percentage of specific findings). Furthermore, patients on immunomodulators and biologics exhibited a progression of the disease activity from mild to moderate GHAS in the follow-up visit. Histological improvement is often delayed in comparison to clinical, laboratory, and endoscopic correction. Overall, this study shows that different therapy modalities impact the progression of CD in various ways.

The disease severity in CD can be measured by the Pediatric Crohn’s Disease Activity Index (PCDAI), and its variants (weighted PCDAI or short PCDAI). We have not been calculating these indices routinely and we could not do retrospective calculations because of a lack of data.

The limitation of our study is a smaller number of patients in the follow-up biopsy category in comparison to the index biopsy group (*n* = 100/45), and the inability to calculate the PCDAI or its variant. The strengths of this study are the inclusion of clinical, laboratory, endoscopic, and histopathologic findings of CD and the subclassification of the results. The PCDAI and the wPCDAI are slightly superior than two other shorter versions and both had comparable correlation with measures of endoscopic inflammation. However, they cannot give a valid assessment of mucosal healing [22].

## 5. Conclusions

Our study demonstrated a significant decline in the frequency of symptoms and an improvement in laboratory values on the follow-up examinations of children with CD. More than a third of our patients had specific endoscopic and histopathologic findings in the upper GI tract, and an additional 23% had HP findings highly suggestive of CD. We demonstrated the importance of regular clinical, laboratory, endoscopic, and histopathological assessments of pediatric CD patients. Additionally, PCDAI, risk factors for disease progression, and dietary data should be included in future work for a better understanding of the complexity of CD. Although the number of follow-up endoscopies and biopsies is increasing, it is still not sufficient. Further prospective studies with larger sample sizes are mandatory and they should include PCDAI or wPCDAI.

## Figures and Tables

**Figure 1 diagnostics-14-00877-f001:**
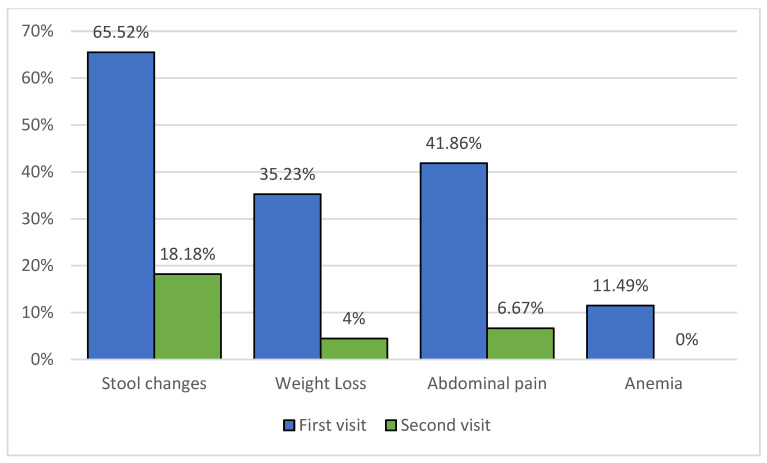
Signs and symptoms on the first and second patient visits.

**Figure 2 diagnostics-14-00877-f002:**
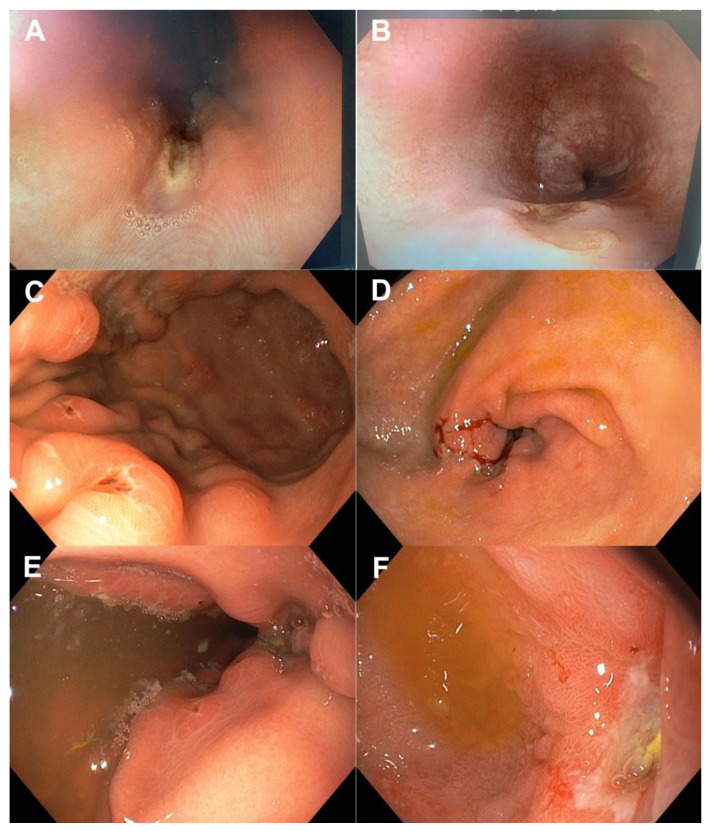
Typical endoscopic features of CD in different localizations: CD oesophagitis (**A**,**B**), CD gastritis (**C**,**D**), and CD duodenitis (**E**,**F**).

**Figure 3 diagnostics-14-00877-f003:**
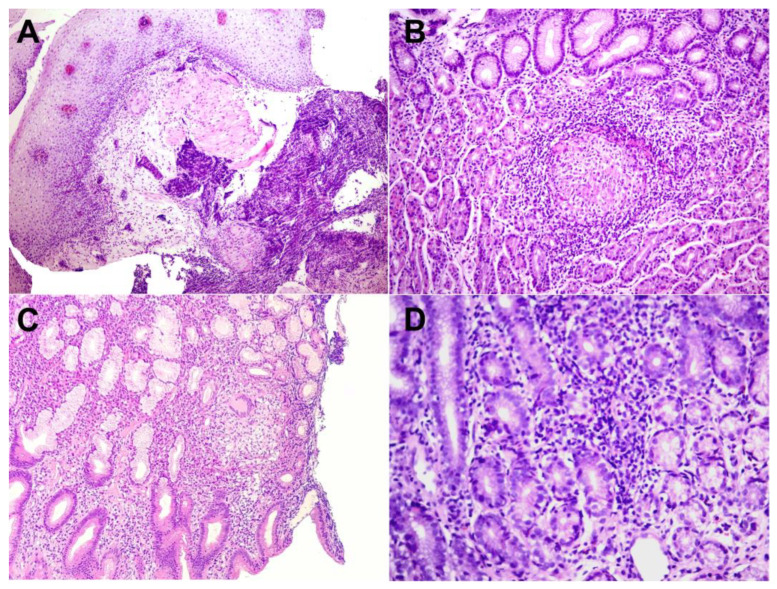
Granulomatous inflammation in CD in the esophageal (**A**): HE, 40×, gastric (**B**): HE, 100×, and duodenal (**C**): HE, 200× mucosa. Focally enhanced gastritis is highly suggestive of CD (**D**): HE, 400×.

**Figure 4 diagnostics-14-00877-f004:**
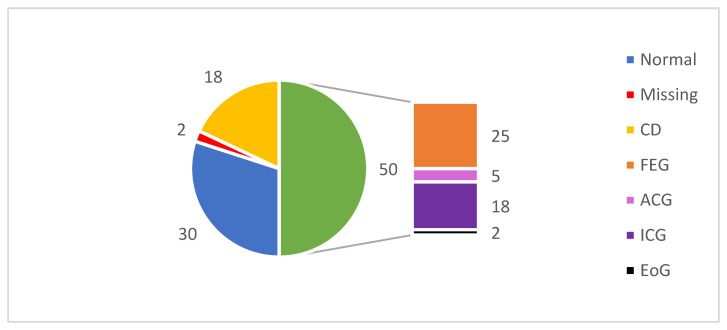
Histopathological findings on gastric biopsies obtained by esophagogastroduodenoscopy. CD (Crohn’s disease); FEG (focally enhanced gastritis); ACG (active chronic gastritis); ICG (inactive chronic gastritis); EoG (elevated number of eosinophils in the gastric mucosa).

**Figure 5 diagnostics-14-00877-f005:**
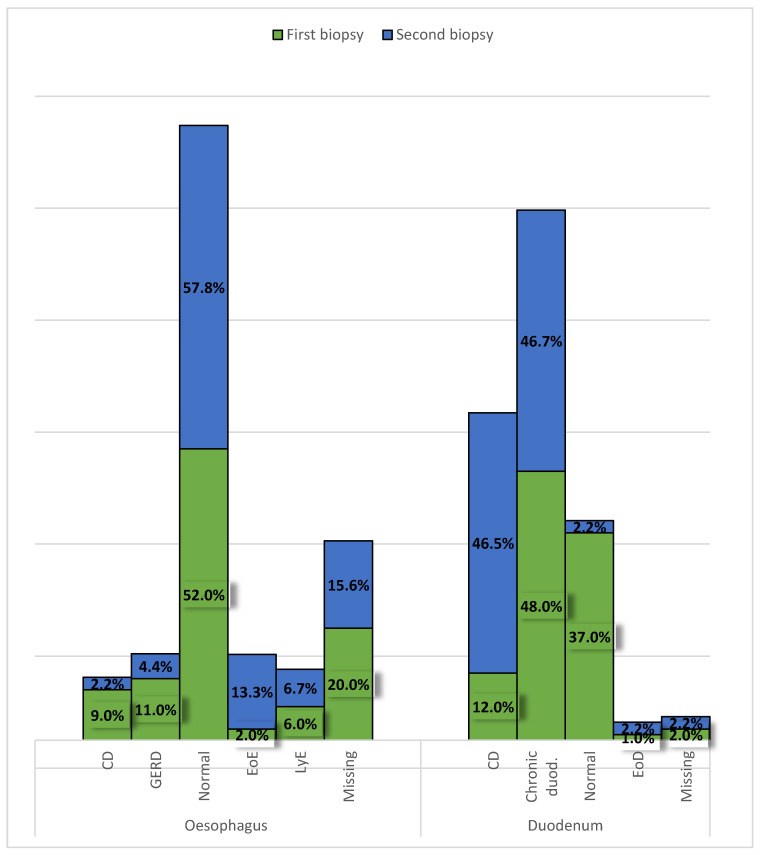
Histopathological findings of the first and second esophageal and duodenal biopsies. CD (Crohn’s disease); GERD (gastroesophageal reflux disease); EoE (eosinophilic esophagitis); LyE (lymphocytic esophagitis)EoD (elevated number of eosinophils in the duodenal mucosa)**.**

**Figure 6 diagnostics-14-00877-f006:**
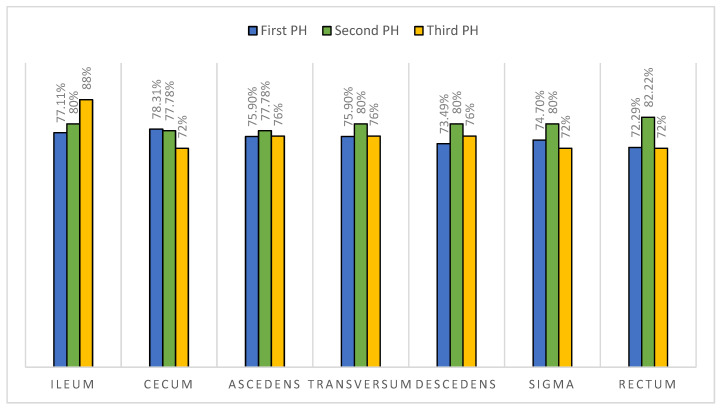
Presence of histopathological findings of CD in the terminal ileum and each colonic segment on the first, second, and third patient visit.

**Table 1 diagnostics-14-00877-t001:** Comparison of macroscopic findings on esophagogastroduodenoscopy (EGDS) and ileo-colonoscopy.

		Ileo-Colonoscopy		
*EGDS*		**Normal**	**Non-Specific**	**Specific**
	Normal	3	9	12
	Non-specific	4	4	10
	Specific	1	2	19

**Table 2 diagnostics-14-00877-t002:** Paris classification of Pediatric Crohn’s disease and GHAS (Global Histology Activity Score).

Paris Classification			GHAS	Disease Activity
Age of onset	A1a	19	Terminal ileum Median 3 (0–14)	None 13 (28.89%) Mild 12 (26.67%) Moderate 14 (31.11%) Severe 6 (13.33%)
	A1b	68		
	A2	13		
Localization	L1	17		
	L2	8		
	L3	68	Colon Median 7 (0–13)	None 1 (1.72%) Mild 21 (36.21%) Moderate 25 (43.1%) Severe 11 (18.96%)
	L4a	32		
Phenotype	B1	98		
	B2	2		
Growth	G0	95		
	G1	5		

## Data Availability

The raw data supporting the conclusions of this article will be made available by the authors on request.

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
