# Peer review of "Pediatric Crohn’s Disease in the Upper Gastrointestinal Tract: Clinical, Laboratory, Endoscopic, and Histopathological Analysis"

_diagnostics, 2024, doi:10.3390/diagnostics14090877_

Round 1

Reviewer 1 Report

Comments and Suggestions for Authors

In this article, the author tracked the endoscopic and histopathological examination results of children and adolescents diagnosed with CD during an 8-year follow-up period, and compared them with clinical symptoms and laboratory indicators. And the distribution of lesions under endoscopy was analyzed. The detailed comments are as follows:

1. In the title, " Pediatric Chron's disease " should be " Pediatric Crohn's disease ". It is recommended that authors carefully check for grammar and spelling issues in the article.

2. In the Intro section, the author summarized the distribution characteristics of pediatric CD, reviewed the histopathological changes corresponding to CD, and pointed out that the standardized grading of pediatric CD has not been established. However, the aim of this article did not mention the corresponding clinical issues. Please further explain the scientific issues that this study focuses on.

3. In the Methods section, it is hoped that the author can provide detailed supplements to the inclusion and exclusion criteria. Meanwhile, the article did not mention the time interval between follow-up visits and the time of receiving treatment for patients.

4. In the Results section, the numbers in Figure 4 were not be clearly displayed.

5. Different pre-treatment severity and treatment regimens may have an impact on changes in endoscopic manifestations and laboratory indicators. Explanation about this content is expected to be supplemented.

Comments on the Quality of English Language

In the title, "Pediatric Chron's disease" should be "Pediatric Crohn's disease". It is recommended that authors carefully check for grammar and spelling issues in the article.

Author Response

Dear Reviewer,

I hope this letter finds you well. I am writing to express my sincere appreciation for your diligent

review. Thank you for investing your precious time in carefully reviewing our work and

providing valuable comments, leading to improvements in the current version. We would like to

assure you that we have thoroughly considered each of your comments and have made every

effort to address them in the current version of the paper. We hope the manuscript, after careful

revisions, meet your high standards. The authors welcome further constructive comments if any.

Below we provide the point-by-point responses. All modifications in the manuscript have been

highlighted (yellow) to facilitate easy identification.

Point 1. In the title, " Pediatric Chron's disease " should be " Pediatric Crohn's disease ". It is recommended that authors carefully check for grammar and spelling issues in the article.

Answer 1. We have corrected the typing mistake in the article title from "Pediatric Chron's disease "to "Pediatric Crohn's disease ". We have reviewed the complete text once more, and corrected all text mistakes.

Point 2. In the Intro section, the author summarized the distribution characteristics of pediatric CD, reviewed the histopathological changes corresponding to CD, and pointed out that the standardized grading of pediatric CD has not been established. However, the aim of this article did not mention the corresponding clinical issues. Please further explain the scientific issues that this study focuses on.

Answer 2. Thank you very much for the suggestion. The aim of the study has been presented in further detail, mentioning clinical, laboratory, endoscopic, and histopathological findings.

Point 3. In the Methods section, it is hoped that the author can provide detailed supplements to the inclusion and exclusion criteria. Meanwhile, the article did not mention the time interval between follow-up visits and the time of receiving treatment for patients.

Answer 3. Thank you for bringing this to our attention. In the Methods section, we have included details about the inclusion and exclusion criteria, and the time interval between follow-up visits. We have also explained that this larger gap between follow-up visits was influenced by the COVID-19 pandemic. The therapy modalities were mentioned, but not quantified.

Point 4. In the Results section, the numbers in Figure 4 were not be clearly displayed.

Anwer 4. In the Results section, the numbers in Figure 4 were corrected.

Point 5. Different pre-treatment severity and treatment regimens may have an impact on changes in endoscopic manifestations and laboratory indicators. Explanation about this content is expected to be supplemented.

Answer 5. Thank you for the valuable suggestion. We have performed the analysis of the impact of different patients’ drug regimens on the laboratory, endoscopic and histopathology results. The three most common regimens (biologics, immunomodulators, and corticosteroids) were matched and compared to the fecal calprotectin, CRP, and albumin levels, endoscopic findings on the upper and lower endoscopy, and GHAS. The results were noted and explained in detail in the discussion segment. For this purpose, we have collected more laboratory results and included them in the study, so that the results may accurately reflect their relationship. We have showed the Paris classification of Pediatric Crohn’s disease and GHAS in a separate table to enhance clarity and readability of the results.

Reviewer 2 Report

Comments and Suggestions for Authors

This is a retrospective single center study included 100 children and adolescents with Crohn’s disease treated at the University Children's Hospital in Belgrade, Serbia, in the period between January 2016 and December 2023.

The authors focus on the endoscopic and histopathological findings in children with diagnosed CD and compare results on the initial and follow-up tests.

This was a very well-done study and the data reported is succinct and well-presented. But still had some scientific issue need further recognized.

1. The title was too similar to previous published article” Schmidt-Sommerfeld, Eberhard, Barbara S. Kirschner, and Janet K. Stephens. "Endoscopic and histologic findings in the upper gastrointestinal tract of children with Crohn's disease." Journal of pediatric gastroenterology and nutrition 11.4 (1990): 448-454.” Authors need re-write their title and avoid similar to published paper.

2. In the Abstract section, there was no numerical data in the results section. Authors should provide more detail results.

3. The rationale is unclear in the introduction section and please improve it.

Author Response

Dear Reviewer,

I hope this letter finds you well. I am writing to express my sincere appreciation for your diligent

review. Thank you for investing your precious time in carefully reviewing our work and

providing valuable comments, leading to improvements in the current version. We would like to

assure you that we have thoroughly considered each of your comments and have made every

effort to address them in the current version of the paper. We hope the manuscript, after careful

revisions, meet your high standards. The authors welcome further constructive comments if any.

Below we provide the point-by-point responses. All modifications in the manuscript have been

highlighted (yellow) to facilitate easy identification.

Point 1. The title was too similar to previous published article” Schmidt-Sommerfeld, Eberhard, Barbara S. Kirschner, and Janet K. Stephens. "Endoscopic and histologic findings in the upper gastrointestinal tract of children with Crohn's disease." Journal of pediatric gastroenterology and nutrition 11.4 (1990): 448-454.” Authors need re-write their title and avoid similar to published paper.

Answer 1. Thank you for bringing this to our attention. We have rewritten the title of our article so that it is not too similar to the previously published article that you have mentioned. According to your recommendation we have changes the title of our article to: “Pediatric Crohn’s Disease in the upper gastrointestinal tract: clinical, laboratory, endoscopic, and histopathological analysis”.

Point 2. In the Abstract section, there was no numerical data in the results section. Authors should provide more detail results.

Answer 2. Thank you for your suggestion. In the Abstract section, we have included more numerical data in the results section, regarding the laboratory and symptoms frequencies.

Point 3. The rationale is unclear in the introduction section and please improve it.

Answer 3.  We have changed the explanation of the aim of the study, as we have analyzed clinical, laboratory, endoscopic, and histopathological findings. Thank you very much for all of your valuable suggestions.

Round 2

Reviewer 1 Report

Comments and Suggestions for Authors

1.The study provides comprehensive analysis by including clinical, laboratory, endoscopic, and histopathological data, allowing for a thorough understanding of CD in pediatric patients. Authors identified specific endoscopic and histopathological findings in the upper GI tract, highlighting the importance of comprehensive evaluation in CD diagnosis and monitoring.

2.But the study has a relatively small sample size, especially in the follow-up biopsy group,and it does not include the calculation of disease activity indices, such as the Pediatric Crohn's Disease Activity Index (PCDAI),which may limit the generalizability of the findings.

3. Discussion of  future directions for research is suggested, which could enhance the overall value of the paper. And how do authors plan to address the limitations mentioned in article in future research?

Comments on the Quality of English Language

1.The study provides comprehensive analysis by including clinical, laboratory, endoscopic, and histopathological data, allowing for a thorough understanding of CD in pediatric patients. Authors identified specific endoscopic and histopathological findings in the upper GI tract, highlighting the importance of comprehensive evaluation in CD diagnosis and monitoring.

2.But the study has a relatively small sample size, especially in the follow-up biopsy group,and it does not include the calculation of disease activity indices, such as the Pediatric Crohn's Disease Activity Index (PCDAI),which may limit the generalizability of the findings.

3. Discussion of  future directions for research is suggested, which could enhance the overall value of the paper. And how do authors plan to address the limitations mentioned in article in future research?

Author Response

Point 1. The study provides comprehensive analysis by including clinical, laboratory, endoscopic, and histopathological data, allowing for a thorough understanding of CD in pediatric patients. Authors identified specific endoscopic and histopathological findings in the upper GI tract, highlighting the importance of comprehensive evaluation in CD diagnosis and monitoring.

 Point 2. But the study has a relatively small sample size, especially in the follow-up biopsy group,and it does not include the calculation of disease activity indices, such as the Pediatric Crohn's Disease Activity Index (PCDAI),which may limit the generalizability of the findings.

Answer 2: Thank you for your valuable cooment. In the previous revised version we have explained that PDCAI and its variants were net calculated because of a lack of data. we clarified that PDCAI and its variants were not calculated due to a lack of data. In the latest version, we have additionally included comments related to PCDAI and its variants, along with an appropriate reference (Lines 380-382 and 452-453). (line 380-382, and 452-453).

Point 3. Discussion of future directions for research is suggested, which could enhance the overall value of the paper. And how do authors plan to address the limitations mentioned in article in future research?

Answers 3:

Thank you very much for this suggestion. We have been added at the end of section Conclusion about our plans for further prospective research in which we aim to overcome the limitations of the current study (line 391-393).

Reviewer 2 Report

Comments and Suggestions for Authors

This was a very well done study and the data reported is succinct and well-presented. Authors had complete response the previous reviewers’ suggestion and revised well. May consider accepted this article.

Author Response

Dear Reviewer, We once again thank you for the valuable suggestions you have provided us with, which have helped us improve our work.
